# Immunosuppressive Therapy and Nutritional Status of Patients after Kidney Transplantation: A Protocol for a Systematic Review

**DOI:** 10.3390/jcm12216955

**Published:** 2023-11-06

**Authors:** Aleksandra Anna Kajdas, Dorota Szostak-Węgierek, Marta Dąbrowska-Bender, Anne Katrine Normann, Ditte Søndergaard Linde

**Affiliations:** 1Department of Clinical Dietetics, Medical University of Warsaw, Erazma Ciolka 27 Street, 01-445 Warsaw, Poland; dorota.szostak-wegierek@wum.edu.pl (D.S.-W.); marta.dabrowska@wum.edu.pl (M.D.-B.); 2Polish Society of Parenteral, Enteral Nutrition and Metabolism (POLSPEN), Banacha 1a Street, 02-097 Warsaw, Poland; 3Department of Gynaecology and Obstetrics, Hospital Southwest Jutland, 6700 Esbjerg, Denmark; anne.katrine.normann.nielsen3@rsyd.dk; 4Department of Clinical Research, University of Southern Denmark, 5230 Odense, Denmark; dsondergaard@health.sdu.dk; 5Department of Gynaecology & Obstetrics, Odense University Hospital, 5000 Odense, Denmark

**Keywords:** immunosuppressive therapy, nutrition, kidney transplant patients

## Abstract

(1) Background: Kidney transplantation is widely recognized as the most effective method of treating end-stage renal disease. Immunosuppressive therapy plays a pivotal role in the treatment of kidney transplant patients, encompassing all patients (except identical twins), and is administered from organ transplantation until the end of its function. The aim of this systematic review is to identify the evidence of the association between immunosuppressive therapy and nutritional status of patients following kidney transplantation. (2) Methods: This protocol has been designed in line with Preferred Items for Systematic Reviews (PRISMA-P). Our search encompasses several databases, including MEDLINE (via PubMed), EMBASE (Elsevier), Scopus and Web of Science. We intend to include observational studies (cross-sectional, case-control, and cohort designs), randomized controlled trials (RCTs), as well as completed and ongoing non-randomized study designs. We will confine our search to studies published in English within the past decade (from inception to 17 February 2023). Qualitative studies, case studies, and conference reports will be excluded. The selection process will be done in Covidence by two independent reviewers. Data extraction will be conducted using a standardized MS Excel template version 16.0. Quality assessment of included studies will be performed using the Cochrane Risk of Bias tool for randomized trials (RoB2), or the Risk of Bias in Non-randomized Studies of Interventions (ROBINS-I) tool. Risk-of-bias plots will be generated using the web application Robvis. Relevant data that have been extracted from eligible studies will be presented in a narrative synthesis. We expect the studies to be too heterogeneous to perform subgroup analyses. (3) Conclusion: This systematic review will offer insights into the evidence regarding association between immunosuppressive therapy and nutritional status of adult patients (18 years of age or older) within the initial year following kidney transplantation. To our knowledge, there is no systematic review addressing that question.

## 1. Introduction

Over the last 50 years, transplantation has achieved remarkable success as a widely adopted medical practice across the globe [1]. Transplanting human tissues, organs, or cells represents a well-established treatment approach recognized as the primary, and often sole, life-saving therapy for numerous severe congenital, hereditary, and acquired diseases and injuries. Transplantation frequently stands as the optimal, and sometimes sole, remedy for both acute and chronic organ failure and it is the only method of treating end-stage heart failure, respiratory system failure, and liver failure. In addition, it is the optimal method of renal replacement therapy [2,3]. Organ transplantation is a complex mechanism. It proceeds in subsequent stages: from determining the possibility of donation, through coordinating its donation, protecting the organ against the consequences of ischemia during the period of its storage, selection of the optimal recipient, transplantation surgery and patient anesthesia, application of an optimal immunosuppressive treatment regimen, and appropriate medical treatment in the postoperative period aimed at preventing or treating complications, as well as early diagnosis of the causes of graft dysfunction. Transplantation, similarly, to many other advanced, expensive medical technologies, should take place only in specialized centers where full laboratory, imaging, and immunological diagnostics can be performed as well as cooperation between doctors of various specialties [3]. Organs can be donated either from a living donor or a deceased donor. It involves an individual agreeing to provide one of their organs to another person, which is then surgically removed and transplanted into the recipient. Organ donation frequency differs from country to country as laws that permit or refuse donation vary [4].

The field of kidney transplantation has also experienced remarkable growth and advancement since Dr. Joseph Murray’s pioneering success with kidney transplantation in 1954 [4]. The most common types of organs transplanted worldwide are kidneys, livers, and hearts, respectively [5]. In 2021, there were an estimated number over 144,000 organ transplants performed globally, showing an increase by 11.3% over the previous year, with kidney transplants accounting for two thirds of these procedures. Most organ transplants take place in Europe and the Americas, primarily due to the greater availability of donors and improved access to these procedures in these regions [5,6]. In addition, the United States, Spain, and Portugal have the highest rates of deceased organ donors worldwide; however, there are still high numbers of patients waiting for organ transplants. In the United States alone there are almost 106,000 candidates waiting for organ donations, the majority of which require a kidney transplantation [4]. It is worth to emphasize that the long-term results of transplantation (preservation of organ function or patient’s survival) depend not only on appropriate management in each of the above-mentioned stages of the transplantation process, but equally on appropriate therapeutic management in the long-term period [3]. The kidneys are a pair of retroperitoneal organs in the body being positioned with the diaphragm above and the 12th rib posteriorly. The right kidney is flanked by the right colonic flexure, liver (hepatorenal ligament), duodenum, and head of the pancreas in front. Conversely, the left kidney is adjacent to the colon’s splenic flexure, the splenic vessels, and the pancreas in the anterosuperior direction. The left kidney also has a connection with the spleen, located anteromedially and linked by the lienorenal ligament. Both kidneys rest on and are in proximity to the psoas muscle medially [5].

Kidney transplantation (kTx) is widely recognized as the most effective method of treating end-stage renal disease [7,8]. It prolongs the patient’s life and significantly improves its quality compared to dialysis [2,4]. Currently, the occurrence of end-stage renal disease is experiencing a rapid increase. The leading causes of it include hypertension and diabetes [9]. When it comes to patients who progress to stage 4 of chronic kidney disease, characterized by a glomerular filtration rate below 30 mL/min/1.73 m, they should be referred to a nephrologist and provided with education regarding kidney failure and various treatment options, including the possibility of kTx [9]. In these cases, as well, kTx offers significantly superior outcomes in terms of survival, quality of life, and cost-effectiveness compared to dialysis [2]. There are two types of kidney donors: living or deceased [4]. The best results for patient and graft survival are obtained in the case of pre-emptive kidney transplantation, which means getting a transplant before starting dialysis from a living donor [7]. Even though kTx is the most effective method of treating ESRD, transplant patients exhibit significant morbidity. They are often burdened with anemia, hypertension, overweight and obesity, diabetes, or cardiovascular diseases [7]. In addition, several complications include: hemorrhage (possible bleeding during or after surgery, often presenting as flank pain or palpable bulges near the incision), thrombosis (rare but serious, with symptoms like hematuria, oliguria, or graft dysfunction), infection (increased risk in the first 3–6 months), arterial Stenosis (late complication, usually asymptomatic, detected by ultrasound), lymphocele (occurs due to lymphatic disruption, causing pain or swelling, treatable with drainage), urinoma (usually within a week post-transplant, causing pain, swelling), or graft dysfunction (confirmed by elevated creatinine) [4]. Variations in the incidence, prevalence, accessibility, and quality of kTx care differs around the globe, with the lowest rates predominantly found in low- and lower-middle income countries. KTx faces several constraints, such as patient eligibility, donor availability, cultural preferences against organs from deceased donors, local or regional medical proficiency, and the expenses associated with kTx surgery and immunosuppressive drugs [9].

The first year for kTx patients is crucial. It is in this period where patients show an increased risk of cancers and infections [4]. Cancer ranks as the second leading cause of mortality among kTx patients. The most frequently occurring cancers in this population are renal cell carcinoma and skin cancer [10], while the most common infections include Epstein-Barr virus (EBV), cytomegalovirus (CMV), and fungal infections. EBV is a human herpesvirus that infects roughly 90% of adults. Among kTx recipients, the most symptomatic infections typically occur with primary infections, which are likely associated with the reactivation of the virus from the donor. One of the most concerning outcomes of EBV infection in this context is the development of post-transplant lymphoproliferative disease. The likelihood of CMV infection is chiefly determined by the CMV serological status of both the donor and recipient. In terms of immunosuppressives, it is important to note that belatacept, which belongs to selective T-cell costimulation blockers, is linked to a heightened risk of primary CMV infection and an extended period of viral replication, particularly in patients at elevated risk of CMV infection [11]. Invasive fungal infections present a substantial risk to kTx recipients. In developing countries, these fungal infections have a notable impact on both patient and graft survival and are the most vulnerable during the early post-surgery period. The most prevalent fungal infections are primarily attributed to fungi within the Candida genus; however, in certain cases, these may be replaced by less harmful infections caused by endemic mycoses like mucormycosis, histoplasmosis, blastomycosis, and coccidioidomycosis [12]. What is more, many kTx patients in the first year following kidney transplantation develop metabolic disorders. Post-transplant diabetes mellitus (PTDM) occurs in 20% of patients within the initial year following kTx and approximately 60% of kTx patients develop lipid disorders, which increase the risk of cardiovascular diseases. Hypertension is also very common among KTx patients. In addition, within the first year after a successful kTx, patients typically increase their body weight by 8–10%, which is considered as a risk factor for dyslipidemia, hyperglycemia, and cardiovascular diseases [13].

The relationship between immunosuppressive therapy, nutritional status, and graft outcomes (such as graft survival and graft rejection) in kTx patients during the first year post-transplant is complex and influenced by multiple factors [13]. First of all, it must be acknowledged that immunosuppressive therapy plays a pivotal role in the treatment of kTx patients. Immunosuppressants must be taken by all (except identical twins) from the organ transplantation until the end of its function [4]. The drugs suppress the recipient’s immune system to shield the transplanted kidney from rejection. It is important to emphasiz that the use of appropriately tailored and consistently administered immunosuppressive therapy is linked to enhanced graft survival [7]. Secondly, the choice of drug and treatment regimen depends on several factors such as prognostic factors depending on the donor and recipient of the organ, etiology of the disease, duration since transplantation (whether it is induction or maintenance), condition of the graft, as well as the interaction of the selected immunosuppressive drug with other drugs used and expected side effects [3,14]. What is important, the cost of treatment should also be considered [3].

There are various immunosuppressive therapies being known among kTx receipients. These strategies have been devised by combining drugs with diverse mechanisms of action effectively. The gold standard immunosuppressive regimen for kTx recipients involves the use of several medications that target various mechanisms and pathways, of which the five main ones include: (1) Calcineurin inhibitors (CNIs) such as cyclosporine and tacrolimus, (2) Mammalian target of rapamycin (mTOR) inhibitors (sirolimus and everolimus), (3) Antiproliferatives like azathioprine and mycophenolic acid derivatives, (4) Glucocorticosteroids, (5) Biological immunosuppressive agents [14]. Contemporary protocols typically involve triple-drug therapy, which includes CNIs, corticosteroids, and antiproliferative drugs. Additionally, mono- or polyclonal antibodies are often employed in induction therapy [15]. To calcineurin inhibitors (CNIs) group we include: cyclosporine A (CsA) and tacrolimus (Tac). They work by inhibiting calcineurin, preventing the activation of T cells, and reducing the immune response. CsA biochemically is a cyclic peptide comprising eleven amino acids. Its metabolism primarily takes place via the cytochrome P (CYP) 450A3 enzyme system, with less metabolism occurring in the gastrointestinal tract and kidneys [14]. Tac seems to be much stronger compared to CsA [16]. It suppresses cellular activity and the humoral immune response through a variety of mechanisms, with the primary impact on calcineurin inhibition [17]. The inhibition occurs by creating a complex with the immunophilin FK 506, which, in turn, hinders the translocation of NF-AT, ultimately suppressing T helper cell proliferation mediated by IL-2 [16]. The bioavailability of Tac varies widely among individual patients, ranging from 5% to 90%. Due to this variability, it becomes challenging to accurately predict the ideal initial dose, make necessary adjustments to the maintenance treatment plan, and assess the potential for adverse effects or treatment ineffectiveness [18]. In the group of mammalian target of rapamycin (mTOR) inhibitors, sirolimus and everolimus are included. These medications impede the proliferation and differentiation of T and B cells, the production of antibodies, as well as the proliferation of non-immune cells, including fibroblasts, endothelial cells, hepatocytes, and smooth muscle cells [19]. Both drugs have similar therapeutic effectiveness [3]. Further, one of the medications classified as an antiproliferative agent is azathioprine. It acts as a purine analogue and exerts its effects by inhibiting the synthesis of nucleic acids. Alongside azathioprine, mycophenolic acid and its two clinically employed derivatives, mycophenolate sodium and mycophenolate mofetil, are among the most utilized immunosuppressives. They play a pivotal role in inhibiting lymphocyte proliferation to prevent the rejection of transplanted kidneys [16,20]. When it comes to glucocorticosteroids, they play a significant role in immunosuppressive therapy to prevent organ rejection. In kTx recipients, two commonly used glucocorticosteroids are methylprednisolone and prednisone. These medications not only have immunosuppressive effects but also possess anti-inflammatory and immunomodulatory properties [16].

While speaking about kindey transplantation it’s not possible not to mention the induction and the maintance therapy. Induction therapy consists of biological agents, including antibodies. They have been created for utilization in induction therapy or for managing transplant rejection. These agents modulate the immune response through various mechanisms. Induction medications can either inhibit lymphocytes or hinder their activation and reproduction, such as the IL-2 receptor antagonist (IL-2RA) [21]. The primary induction drugs used in kTx include basiliximab, alemtuzumab, and antithymocyte globulin [22]. Basiliximab is a chimeric monoclonal antibody that specifically targets the alpha chain of the interleukin-2 receptor (IL-2R). It functions by binding to and obstructing this alpha chain, thereby inhibiting the activation of IL-2R. Alemtuzumab, on the other hand, facilitates the breakdown of T and B lymphocytes, monocytes, and NK cells in the peripheral blood. This process results in a significant and long-lasting reduction in T lymphocytes, along with a more temporary reduction in B lymphocytes and monocytes [16]. One of the antibodies mentioned is antithymocyte globulin, which is generated by immunizing animals with human lymphoid cells. The most effective antibodies among these can recognize a range of markers including CD2, CD3, CD4, CD8, CD11a, CD18, CD25, CD28, CD40, and CD54. These antibodies exhibit a broad immunosuppressive spectrum compared to monoclonal antibodies [16]. Additional biologic drugs, such as eculizumab, belatacept, and rituximab, are frequently employed in specific clinical situations [23]. The choice of induction therapy depends on patient’s risk assessment, immunological condition, and the specific protocols followed by the transplant center. Following the induction phase, the maintenance therapy is continued. It is to ensure long-term graft survival and minimize the risk of side effects and complications connected to immunosuppression. Frequently used immunosuppressive drugs in the maintenance therapy of kTx recipients include CNIs, mTOR inhibitors, antiproliferative agents, and corticosteroids [24].

Although all immunosuppressive therapies have a positive effect on kTx patients’ outcomes and graft survival, they may present several side effects. For example, they can exacerbate pre-existing metabolic disorders like lipids or carbohydrates and induce new ones such as hypophosphatemia or hypomagnesemia [7]. Furthermore, immunosuppressives elevate the risk of obesity, infections, anemia, diabetes, and bone diseases, which can indirectly impact graft outcomes [8]. CNIs can lead to disruptions in multiple bodily systems, affecting the urinary system, gastrointestinal tract, and circulatory system [25]. In addition, the most frequently observed side effects associated with mTOR inhibitors include pneumonitis, thrombotic microangiopathy, complications in the healing of surgical scars or infections, lymphocele, excessive surgical drainage, post-transplant diabetes mellitus, hypertriglyceridemia, hyperlipidemia proteinuria, and oedema [26]. Due to significant complications among kTx patients regarding PTDM, hypertension, and high cholesterol levels, it is advisable to minimize the use of glucocorticoids whenever possible. Additionally, addressing and preventing anemia is a crucial consideration. In this context, the treatment of kTx recipients can encompass approaches such as vitamin D supplementation, folic acid, vitamin B6 and B12 treatment, and the use of anti-inflammatory and antioxidant drugs. Moreover, a healthy lifestyle, which includes quitting smoking, adopting a nutritious diet, and, most importantly, engaging in regular physical activity, plays a pivotal role in reducing cardiovascular risk [27].

When it comes to nutrition, it is widely acknowledged that the accurate nutritional status of patients plays a significant role in post-transplant results [28]. It is well-documented that proper nutritional status of kTx patients can enhance the outcome of a successful transplant and improve patients’ survival. Additionally, it serves as an important indicator of an individual’s overall health and quality of their life [29]. As the opposite, inadequate nutrition or a suboptimal nutritional status can compromise the immune system’s function, potentially leading to complications and graft rejection. What is more, excessive weight gain or obesity can heighten the risk of metabolic complications and cardiovascular issues, which can indirectly influence the condition of the graft. It is essential to consider individual patient characteristics, existing medical conditions, and variations in immunosuppressive treatment plans when evaluating these interconnections [30]. The nutritional management of kTx patients can be divided into four periods: the pre-transplant period, the perioperative period, the early and the late post-transplant periods. Post-transplant nutrition aims to conserve protein resources despite increased catabolism, support proper wound healing, and prevent infections and electrolyte imbalances [7].

Thankfully, the nutritional status is becoming an integral part of the clinical evaluation of individuals with compromised organ systems. To determine patients’ nutritional status, calculations of biochemical and anthropometric indexes, immunological tests, as well as subjective assessments based on clinical examination are now being used [31,32]. The use of traditional anthropometric indexes is subject to doubt due to inherent limitations in the technique. These limitations include poor reproducibility, both between different examiners and within the same examiner, variability in the calibration of skinfold calipers, inconsistencies in identifying the precise measurement site, and the potential influence of generalized edema, all of which can hinder the accurate application of anthropometry. Bioelectrical Impedance Analysis (BIA) is a noninvasive and convenient method used at the bedside to evaluate the nutritional status of individuals. It is anticipated that patients who have undergone a successful kTx will exhibit favorable nutritional intake, and over time, they will gradually regain their overall health, resulting in a positive nutritional status [30].

There is some understanding of the impact of immunosuppressive therapy on the nutritional status of patients within the initial year following kTx. Researchers examine various outcomes such as weight, the prevalence of diabetes, lipid abnormalities, or deficiencies in minerals and vitamins [33,34,35,36]. For example, in a cross-sectional study, vitamin B12 deficiency was linked to reduced dietary intake of B12 as well as the use of mycophenolate mofetil [33]. However, in a clinical trial, cyclosporine was associated with increased HDL and LDL cholesterol levels at 1-month post-transplant. In contrast, hyperlipidemia was less pronounced in patients receiving tacrolimus [34]. In a cohort of 70 kTx patients, excess body weight was associated with various factors, including immunosuppressive therapy [35]. Another study based on 3342 participants identified higher cyclosporine levels as risk factors for hyperglycemia [36]. Immunosuppressives that contribute to development of anemia among kTx recipients may include mycophenolate mofetil, Tac, azathioprine, mTOR inhibitors, blockers of the renin-angiotensin-aldosterone system angiotensin-converting enzyme inhibitors and angiotensin receptor antagonists, allopurinol, and trimethoprim. [37,38]. Hypovitaminosis D is a commonly observed condition following kTx [39,40]. Higher prednisone doses are correlated with decreased vitamin D concentrations and an increased likelihood of deficiency. This connection can possibly be explained by the stimulatory impact of glucocorticoids on the breakdown of vitamin D. This could also be associated with the underlying reason necessitating the higher steroid dosage. The use of mycophenolate sodium, regardless of the dosage, was linked to vitamin D deficiency. In addition, patients prescribed higher doses of tacrolimus showed an increased likelihood of developing vitamin D deficiency [41]. A notable impact of calcineurin inhibitor (CNI) intake on vitamin D was noted [42].

In kTx recipients, the range of bone-related conditions encompasses renal osteodystrophy, osteoporosis, bone fractures, and osteonecrosis [43]. Previous studies conducted after transplantation suggest that bone mineral density (BMD) experiences an initial decline of 4% to 10% during the first 6 months. Furthermore, there is an additional decrease in lumbar BMD of approximately 0.4% to 4.5% in the period between 6 and 12 months [44,45]. The weakening of bone structure results in the use of steroid drugs and reduced calcium absorption [13,43]. Nonetheless, the heightened risk of bone loss after transplantation remains even in the absence of steroids. For instance, in a recent study involving 47 kidney transplant recipients who had their glucocorticoids (GCs) discontinued just three days after transplantation, there was a notable decline in bone mineral density (BMD) at the distal radius after 12 months, even though BMD at the lumbar spine and hip did not decrease [46].

In a minority of cases, PTDM can potentially be reversed by modifying the immunosuppressive treatment (such as reducing steroid doses or switching from tacrolimus to cyclosporine), adhering to a suitable diet, and maintaining an optimal body weight [13]. A substantial retrospective study encompassing over 25,000 transplant recipients revealed that immunosuppression without steroids was linked to a lower risk of post-transplant diabetes mellitus (PTDM) compared to regimens containing steroids. The cumulative incidence of PTDM within 3 years after transplantation was 12.3% for steroid-free therapy and 17.7% for therapy containing steroids [47]. However, in a recent double-blind study lasting 5 years, which compared a group of patients who initially discontinued corticosteroids with another group that reduced corticosteroid dosage to 5 mg per day after 6 months, no significant difference was observed in the PTDM rate. The PTDM rate was 35.9% for the group that stopped corticosteroids and 36.3% for the group that maintained a 5 mg daily dosage after 6 months [48]. Glucocorticoids induce hyperglycemia through various mechanisms, including heightened insulin resistance, reduced insulin secretion, the initiation of beta cell apoptosis, and decreased expression of glucose transporters [36,49,50,51,52]. Lipid disorders can be attributed to various factors, including the use of immunosuppressive drugs, such as steroids, CNIs (with a preference for cyclosporine over tacrolimus), and mTOR inhibitors like sirolimus and everolimus [53]. Arterial hypertension is highly prevalent among patients following kidney transplantation and serves as a risk factor for cardiovascular diseases [54]. Weight gain occurs in most patients and results in the development of overweight and obesity. In many cases, these are overweight and obesity combined with sarcopenia. The term sarcopenia characterizes loss of function associated with loss of skeletal muscle mass. This condition is referred to as presarcopenia when only loss of muscle mass is detected. Its etiology is multifactorial, but sarcopenia is usually the result of chronic diseases, immunosuppressive treatment, malnutrition, or lack of exercise, which leads to muscle atrophy. Steroid drugs, lack of physical activity, and improper diet contribute to a reduction in the lean tissue index [35,55,56,57]. Hypophosphatemia is a frequent occurrence both in the early and late stages following kTx. It is linked to the development of bone-related complications, including conditions like osteomalacia and osteodystrophy. Hypomagnesemia is a common finding in most patients after kTx, particularly during the initial period. The use of calcineurin inhibitors, notably tacrolimus, tends to contribute to its occurrence [58]. What is more, patients who have undergone kTx may experience fluctuations in serum potassium levels, including both low and high values. Dietary recommendations should be tailored based on the results of laboratory tests to ensure appropriate management. What is more, immunosuppressive therapy can impact vitamin B6 metabolism in individuals who have undergone kidney transplantation [59]. The use of immunosuppressive drugs, including corticosteroids, cyclosporine, tacrolimus, and sirolimus, can potentially worsen the progression of diabetes, hypertension, and hyperlipidemia [60].

The aim of proposed systematic review is to identify the evidence of the association between immunosuppressive therapy and nutritional status of patients following kidney transplantation.

## 2. Materials and Methods

### 2.1. Protocol and Registration

This protocol has been designed in line with the Preferred Reporting Items for Systematic Reviews and Meta-Analyses Protocols (PRISMA-P) checklist 2015 [61] [Appendix A]. The protocol was registered at the International Prospective Register for Systematic Reviews (PROSPERO) prior to study conduct [CRD42023396773] (registration date: 12 April 2023) [62]. This systematic review will be reported according to the Preferred Reporting Items for a Systematic Reviews and Meta-Analyses (PRISMA) 2020 checklist [63] [Appendix A]. Visual representation of the systematic review development is presented in Figure 1.

### 2.2. Eligibility Criteria

The studies will be included based on the following criteria: population, exposure, and outcome.

#### 2.2.1. Types of Participants

We will include studies considering adult patients (18 years of age or older) within the initial year post kidney transplantation. We will exclude studies based on children, young adults, and pregnant women. We will include studies only based on kidney transplantation. Therefore, patients who have undergone multiple organ transplants will be excluded.

#### 2.2.2. Exposure(s)

Our exposure will include different schemes of immunosuppressive therapy, including various types and doses of immunosuppressive drugs (i.e., Tacrolimus, Cyclosporine, glucocorticoids). We will include only studies where the type of immunosuppression therapy used for the patients is stated. In addition, we will include only studies that check the effect of the immunosuppression on various nutritional outcomes.

#### 2.2.3. Outcomes(s)

Our primary outcome will be the prevalence, incidence, frequency, and 95% cumulative incidence (CI) of various nutritional characteristics such as weight, body mass index (BMI), glucose level, vitamins (i.e., Vitamin B6, Vitamin B12, and Folic Acid (also known as Vitamin B9)), as well as minerals levels (i.e., Iron (Fe), Magnesium (Mg), Phosphorus (P), and Potassium (K)). In addition, bioimpedance (BIA) analysis factors (such as, i.e., per cent body fat, mass of body fat, lean body mass, total body water, body cell mass, skeletal muscle mass) will be considered.

### 2.3. Information Sources

We will search the following databases: MEDLINE (via PubMed), EMBASE (Elsevier), Scopus, and Web of Science. We will include observational studies (cross-sectional, case-control, and cohort), randomized controlled trials (RCTs), and non-randomized study designs, both completed and ongoing, published in English within the last 10 years (from inception till 17 February 2023). We will exclude qualitative studies, case studies, conference’s reports, as well as literature reviews and studies based on children. There is no limitation in any setting. The reference lists of the articles relevant to our research question will be searched additionally. Where data is missing, the authors will be contacted.

### 2.4. Search Strategy

A comprehensive search strategy was developed in collaboration with an experienced research librarian from the Medical University of Warsaw and is presented as Appendix A. To construct accurate search terms, we used subject headings and subheadings as well as text words that will be used to describe words and phrases. For example, in the MEDLINE (via PubMed) database for ‘kidney transplantation’, we used the term “kidney transplantation” found in [All fields], as well as “organ transplantation”, “Renal Replacement Therapy”, and “Transplants”, all found as MeSH terms; for ‘immunosuppression therapy’: “immunosuppression therapy” [MeSH Terms], ((“Immunosuppressive Agents”), “Immunosuppressive Agents” [MeSH Terms], “Immunosuppressive scheme”, “immunocompromised host” [MeSH Terms]; and for nutritional status, “nutritional status”, “body composition”, “body composition” [MeSH Terms], “body mass index”, “body mass index”[MeSH Terms] etc. Each group has been combined using operators AND, OR, and NOT. According to each database guidelines, EMBASE (Elsevier), Scopus, and Web of Science, we applied all the rules in our search strategy. The search was revised and approved by all authors.

### 2.5. Study Selection

Selection of studies will be done in Covidence, which is a web-based systematic review management tool [64]. After removing duplicates, two authors (AK, AKN) will blindly screen titles and abstracts for obvious exclusion followed by full-text screening. Disagreements will be solved through discussion, or with help of a third independent reviewer (DSL), if needed. Selection of studies will be presented using a flow chart based on the PRIMSA flow chart 2020 [Appendix A].

### 2.6. Data Extraction

Data extraction and management of data will be done by two reviewers (AB, AKN) and will be exported to the standardized MS Excel template version 16.0. Any disagreements will be solved through discussion, or an inclusion of a third reviewer (DSL), if needed. The following data will be extracted: (1) study reference (first author and year of publication), (2) study design, (3) country where the study was conducted, (4) settings, (5) sample size, (6) mean age of participants, (7) number of males and females, (8) description of exposure, (9) description of outcome, (10) measures of effects and tools of measurements, and (11) how the data was gathered (from medical records, colleting blood samples, etc.), (12) results, (13) conclusion, and (14) quality of studies included. We will also include limitations (response bias, selection bias, limitations of assessment tools, information bias).

### 2.7. Quality Assessment

Two investigators (AK and MBD) will assess the bias of eligible RCT’s studies using the Cochrane Risk of Bias tool (RoB2, August 2019) [Appendix A] [65]. That includes, as follows, sequence number generation, concealment of allocation sequence, blinding, incomplete outcome, selective outcomes, using the Review Manager (RevMan) [66]. Two investigators (AK and AKN) will perform bias assessment using the Risk of Bias in Non-randomized Studies of Interventions (ROBINS-I) tool [67] [Appendix A]. Initially, the risk of bias will be evaluated in the seven subdomains of ROBINS-I, and subsequently, an overall risk of bias judgment will be made. Risk-of-bias plots will be generated using the web application Robvis [68].

For the evaluation of the evidence produced in this systematic review, we will use the GRADE approach. We will use methods and recommendations described in Section 8.5 and Chapter 12 the Cochrane Handbook of Systematic Reviews of Interventions using GRADE software, version updated on January 2022 [69].

### 2.8. Data Analysis

The results of our study search, conducted on scientific databases such as MEDLINE (via PubMed), EMBASE (Elsevier), Scopus, and Web of Science, will be visually presented in a Flow Diagram for the year 2020.

Relevant data that have been extracted from eligible studies will be presented in a narrative synthesis. Data will be presented in tables that characterize all studies included in our review. The aspects that will be presented include: study reference (first author and year of publication), study design, a country where the study was conducted, settings, sample size, mean age of participants, number of males and females, description of exposure, description of outcome, measures of effects and tools measurements, and how the data was gathered (from medical records, colleting blood samples etc.), results, conclusion, and quality of studies. As our exposure, we consider different types and doses of immunosuppressives; in each article, we will check the specific immunosuppressives used. We expect outcomes such as vitamins deficiencies, occurrence of diabetes, or weight among patients in the initial year following kidney transplantation. We will group descriptively studies based on the same exposure and/or the same outcome and will draw conclusions based on that. In addition, we will also consider a study design and other relevant aspects, where information will be provided. In a narrative synthesis, we will discuss how potential confounders are addressed or controlled for in each study.

We anticipate that the studies may exhibit significant heterogeneity in terms of methodology, types of population, and intervention, making it challenging to conduct subgroup analyses. Nevertheless, if we identify eligible studies that demonstrate homogeneity concerning participants, exposure, outcome, and study design, we will proceed with a meta-analysis. In such instances, a statistician will be enlisted to assist in the data analysis. SAS 9.4 will be used for statistical analysis. Meta-analysis will be done using Rev-Man 5.3 (Cochrane). We plan to use both continuous and dichotomous outcomes data. If trials present continuous data with a consistent outcome measurement scale, we will conduct an analysis using the mean difference. In cases where trials employed diverse scales for their data, we will perform an analysis using the standardized mean difference and will compute the associated 95% confidence intervals. Statistical heterogeneity will be assessed using I^2^. If trials include multiple time points for follow-up, we will utilize the most recent time point for our analyses. We intend to carry out subgroup analyses to compare trials with an overall low risk of bias to those with a high risk of bias. The Egger’s test will be used to evaluate the potential presence of publication bias in a meta-analysis by examining the asymmetry in the funnel plot. For the evaluation of the evidence produced in this systematic review, we will use the GRADE approach. We will use methods and recommendations described in Section 8.5 and Chapter 12 the Cochrane Handbook of Systematic Reviews of Interventions using GRADE software, version updated on January 2022. Sensitivity analysis will be conducted to explore the impact of study sample sizes on the overall findings. It will be checked if excluding smaller or less precise studies influences the results. When conducting a meta-analysis, the impact of confounding in each study will be considered. If there is sufficient data and heterogeneity across studies, subgroup analyses based on different levels of confounders or other study characteristics will be considered. We will summarize and compare results using statistical (meta-analytic) methods, examining various endpoints. These endpoints will encompass factors such as disease incidence, prevalence, or cumulative risks in relation to the nutritional status of our participants. Throughout our analysis, we will condense, juxtapose, and scrutinize the characteristics and findings of the studies, delving into potential relationships between them. For a better understanding of the concepts, we will perform graphs, figures, and tables of the presented results.

## 3. Conclusions

Kidney failure represents a significant public health concern. Its impact is expected to surge significantly due to the aging of the general population and the increasing prevalence of conditions such as diabetes and hypertension. While dialysis remains the prevailing treatment for kidney failure across the globe, research conducted in both high-income and middle-income countries has indicated that kidney transplantation (kTx) is a more cost-effective option. Additionally, kTx is associated with improved quality of life, enhanced survival rates, and greater economic productivity [9]. Given that the nutritional status of kidney transplant patients holds critical implications for their health and graft survival, and recognizing the inevitable influence of necessary immunosuppressive therapy on patients’ nutritional status and overall health, coupled with the global rise in kidney transplants, we intend to conduct this systematic review.

To the best of our knowledge, this will be the first systematic review aimed at identifying the evidence regarding the association between immunosuppressive therapy and the nutritional status of patients after kidney transplantation. The anticipated benefits of this research encompass: (a) fostering a deeper comprehension of the intricate relationship between the employment of immunosuppressive drugs—encompassing their types, regimens, and dosages—and the nutritional status of patients within the initial year post kidney transplantation; (b) discerning fluctuations in the nutritional status of kidney transplant recipients, both in the immediate aftermath and over a prolonged period following kidney transplantation; (c) delineating risk factors associated with the use of immunosuppressants. In our systematic review, we hold a keen interest in delving into the realms of nutritional status and immunosuppressive therapy among kTx patients, for several compelling reasons: (1) The adequacy of nutrition in kTx patients often serves as a barometer for the success of the transplanted graft and the patients’ survival; (2) Proper nutrition emerges as a pivotal gauge for an individual’s overall health and quality of life; (3) Considering metabolic disorders like overweight and obesity, hypertension, or anemia, kTx may introduce a risk factor for potential future adverse outcomes; (4) The impact of immunosuppressive therapy on the nutritional status of kTx patients can lead to severe health issues, including disorders in lipids and carbohydrates, hypomagnesemia, hypophosphatemia, bone diseases, and anemia [10]. We have opted to adopt various viewpoints to underscore the significance of our systematic review. Firstly, by adopting a public health perspective, we aim to identify patients who may necessitate nutritional interventions post-kTx. It is imperative to recognize that nutritional interventions are not only cost-effective but also pivotal to consider within the framework of kTx treatment and assessment. Delving into prevailing trends in immunosuppressive therapy and its impact on the nutritional status of post-kTx patients holds profound clinical implications. This undertaking will offer insights critical to shaping policies governing post-kTx patient care, a vantage point of paramount importance from a policy perspective. Additionally, we intend to bridge a research gap by comprehensively addressing the nutritional status of kTx patients alongside immunosuppressive therapy, particularly through a more extended follow-up period that aggregates pertinent studies for a holistic perspective.

This systematic review will be conducted using explicit and rigorous methods, and led by experienced researchers, which is its strength. However, we acknowledge certain limitations. Our inclusion criteria encompass studies only in English and are subject to time restrictions which are among the limitations of this systematic review.

Ethics approval is not deemed necessary for a systematic review. The outcomes of the planned systematic review will be disseminated through publication in an open-access journal indexed for broader accessibility. Additionally, the findings will be presented at relevant conferences.

## Figures and Tables

**Figure 1 jcm-12-06955-f001:**
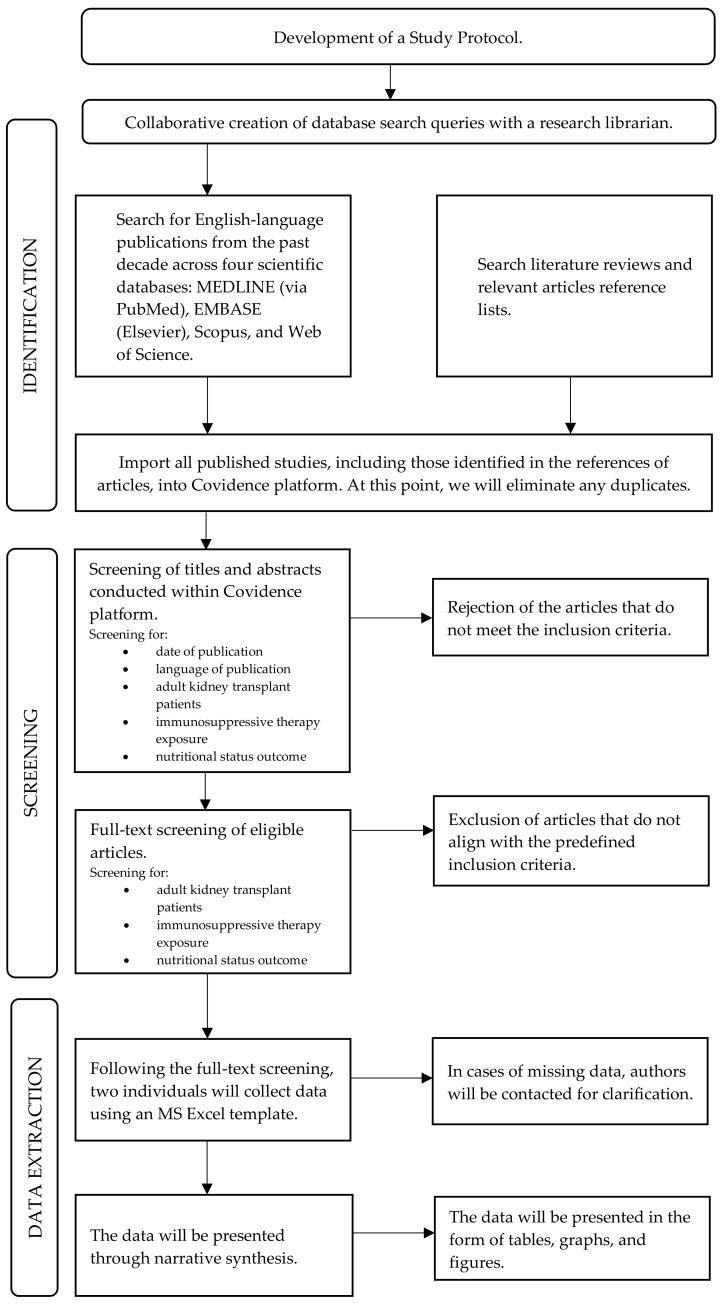
Visual representation of the systematic review development.

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
