# Peer review of "Immunosuppressive Therapy and Nutritional Status of Patients after Kidney Transplantation: A Protocol for a Systematic Review"

_jcm, 2023, doi:10.3390/jcm12216955_

Round 1

Reviewer 1 Report

Comments and Suggestions for Authors

Authors intend to study relationship between immunosuppressive therapy and nutritional status of the transplant recipients. I have following comments to make:

More details of statistical analysis needed especially following questions need to be answered:

How will be the heterogeneity assessed?

How will be the effect size assessed?

How will be confounders taken care of?

Which software authors are planning to use for statistical analysis?

How will be the publication bias and small sample size issues handled?

The introduction is way too long and irrelevant. Authors should concentrate on the work they intend to do. So, it should include the relevant discussion on transplantation, various immunosuppressants including induction, individual effects that these can have on the nutritional outcomes that the authors are intending to study and what are the pathophysiological mechanism for the same. There is no point in discussing CKD classification, various RRT etc. in the introduction.

There is no need of discussion as the study is not done yet. It is just a protocol for the systematic review. The authors need to write a good relevant introduction, discuss protocol and end it with conclusion.

Avoid repetition of similar sentences. Keep it brief and crisp.

Article needs to be proof-checked thoroughly

Comments on the Quality of English Language

mentioned in the comments file

Author Response

Dear Sir/Madame,

thank you for your time and valuable comments regarding our Protocol for a Systematic Review. We have referred to all of your suggestions and have tried to answer them in details point-by-point. In addition, we have tried to apply them in the revised version of the manuscript submitted. Please see attachment below. 

Kind regards, 

Aleksandra Kajdas

Reviewer 2 Report

Comments and Suggestions for Authors

This is a well-designed study/review protocol. However, the reviewer wonder where the result session of this review paper was.

What are the association between immunosuppressive therapy/nutritional status and the graft status (survival, graft rejection etc.) among the KTx patients during the first-year post-transplant?

Author Response

(The authors gave the same response as above.)

Round 2

Reviewer 1 Report

Comments and Suggestions for Authors

Acceptable

Comments on the Quality of English Language

Acceptable